# Retention in Care and Virological Failure among Adult HIV-Positive Patients on First-Line Antiretroviral Treatment in Maputo, Mozambique

**DOI:** 10.3390/v15101978

**Published:** 2023-09-22

**Authors:** Leonid Joaquim, Mafalda N. S. Miranda, Victor Pimentel, Maria do Rosario Oliveira Martins, Tacilta Nhampossa, Ana Abecasis, Marta Pingarilho

**Affiliations:** 1Centro Integrado de Cuidados e Tratamento, Hospital Militar de Maputo, Maputo P.O. Box 21414368/9, Mozambique; 2Global Health and Tropical Medicine (GHTM), Associate Laboratory in Translation and Innovation Towards Global Health (LA-REAL), Instituto de Higiene e Medicina Tropical (IHMT), Universidade NOVA de Lisboa (UNL), Rua da Junqueira 100, 1349-008 Lisboa, Portugal; mafaldansmiranda@ihmt.unl.pt (M.N.S.M.); victor.pimentel@ihmt.unl.pt (V.P.); mrfom@ihmt.unl.pt (M.d.R.O.M.); ana.abecasis@ihmt.unl.pt (A.A.); martapingarilho@ihmt.unl.pt (M.P.); 3Centro de Investigação em Saúde de Manhiça, Sede, Manhiça, Rua 12, Maputo 1929, Mozambique; tacilta.nhampossa@manhica.net

**Keywords:** HIV-1, retention on care, virological response, Mozambique

## Abstract

Introduction: Access to antiretroviral treatment (ART) is increasingly available worldwide; however, the number of patients lost to follow-up and number of treatment failures continue to challenge most African countries. Objectives: To analyse the retention in clinical care and the virological response and their associated factors of HIV-1 patients from the Maputo Military Hospital (MMH). Methods: A cross-sectional observational study was conducted to analyse data from patients who started ART between 2016 and 2018 in the MMH. Results: At the end of 12 months, 75.1% of 1247 patients were active on clinical follow-up and 16.8% had suspected virologic failure (VL > 1000 copies/mm^3^). Patients younger than 40 years old were more likely to be lost to follow-up when compared to those aged >50 years old, as well as patients who were unemployed and patients with a CD4 count < 350 cells/mm^3^. Patients with haemoglobin levels lower than 10 g/dL and with a CD4 count < 350 cells/mm^3^ were more likely to have virological failure. Conclusions: We have identified clinical and sociodemographic determinants of loss to follow-up and in the development of virological failure for HIV-positive patients in clinical care in the MMH. Therefore, HIV programs must consider these factors to increase the screening of patients at high risk of poor outcomes and particularly to strengthen adherence counselling programs.

## 1. Introduction

Human immunodeficiency virus (HIV) is still one of the major public health problems worldwide, especially in low–mid-income countries (LMIC). It is estimated that in 2021, worldwide, around 38.4 million individuals were living with HIV [1], and 650.000 deaths from AIDS-related illnesses were registered in the same year [2]. Also in 2021, in Africa, it was estimated that around 25.4 million people were infected with HIV, with 470.000 deaths having been registered in 2018 [3]. Due to the use of antiretroviral therapy (ART), HIV became a chronic disease, and disease progression can be controlled through the lifelong use of ART [4].

In Mozambique, in 2021, the estimate of inhabitants living with HIV was 2.101 million, with around 35.463 deaths recorded from HIV [5]. Also in 2021, Mozambique reported an average of 94.053 new infections, corresponding to 258 new cases per day [5]. The peak of HIV prevalence was registered among the age groups between 35 and 39 years, within which a higher prevalence was observed in females (26.6%) compared to males (15.9%) [6].

To address the pandemic, the World Health Organization (WHO) with the Joint United Nations Program on HIV/AIDS (UNAIDS) launched the 95-95-95 targets to be achieved in 2030, which propose achieving 95% diagnosis among all people living with HIV (PLHIV), and of those, 95% of patients on antiretroviral therapy (ART), and of those, 95% patients should achieve viral suppression (VS) [7].

Mozambique has adopted this 95-95-95 strategy to improve the access to treatment for HIV-positive people. However, the massive expansion of units offering ART has become challenging, making it difficult to keep the balance between ART demand and availability, as well as the quality of clinical care [8]. In 2021, of the 1.706 existing health facilities in Mozambique, around 96% offered ART services [5]. Of the 2.1 million people estimated to be living with HIV in this country, 84% were diagnosed, 81% of those were on ART, and of those, only 71% had achieved viral suppression [5]. Virological suppression has been a great challenge for the Ministry of Health in Mozambique due to the low levels of retention in clinical care and weak ART adherence. In fact, data from 2018 showed that at 12 months of ART, the retention was 68% [9].

Several studies have shown evidence that with an adequate ART regimen, an increase in survival and an improvement in the quality of life of these patients is observed [10]. However, several sociodemographic (such as age and gender) and clinical factors contribute to poor retention in care and virological response, which also contributes to an increase in new HIV infections. Thus, this study aims to analyse the retention in care and the virological response and their associated factors of HIV-1 patients from the Maputo Military Hospital (MMH).

## 2. Methods

This is a cross-sectional observational study conducted at the Maputo Military Hospital (MMH) in Mozambique. The MMH belongs to Mozambique’s national health system and is the largest military health facility in the country. It has an Integrated Care and Treatment Centre (CICTRA), where ART services are offered, with about 16,244 HIV+ patients registered in its database (openMRS). The follow-up of HIV-positive patients started in 2005 and the test and treatment strategy was adopted in 2016. It has an average daily observation of 90 HIV-1 patients and an average annual ART initiation of 780 patients.

### 2.1. Study Population

All patients who started treatment at the HMM between the 1st of January 2016 and 30th of December 2018 and who were older than 18 years old were selected from the HMM database (openMRS). Pregnant and lactating women, patients co-infected with tuberculosis and deceased patients were excluded from this study. Moreover, those who were transferred to and from other health facilities during the period of analysis and from whom clinical information and follow-up were lost were also excluded from this study. Overall, 1773 newly diagnosed patients were retrieved from the openMRS database and 1247 were included in the analysis, according to the flowchart presented below (Figure 1).

### 2.2. Definitions

The Ministry of Health of Mozambique defines virological failure as patients with a viral load above 1000 copies/mm^3^ and good virological response as patients with a viral load below 1000 copies/mm^3^ 6 months after starting ART [11]. The level of CD4 values was stratified into two categories, distinguishing those who started treatment with CD4 < 350 cells/mm^3^ (late presenters) and CD4 ≥ 350 cells/mm^3^ (non-late presenters). The haemoglobin level was stratified into ≥10 g/dL and <10 g/dL [12].

### 2.3. Variables and Measurements

The two main outcomes were retention in care at 12 months and virological failure and the independent variables were sex, age at the start of treatment, distance between residence and health care facility, employment, education, WHO clinical stage, body mass index, haemoglobin, CD4 count and ART regimen.

### 2.4. Management and Statistical Analysis of Data

Data were entered, cleaned, coded and checked for missing values, outliers and inconsistencies.

Descriptive analysis was used to characterize the sociodemographic and clinical characteristics of patients included in this study.

To analyse the association between the outcome and qualitative variables, chi-square tests (χ^2^) or Fisher tests were used. To compare age at the start of the treatment between the two groups, the Mann–Whitney test was used. 

To examine the association between sociodemographic and clinical factors and the occurrence of dropouts of follow-up and virological failure, a multivariable analysis was used, using logistic regression together with an adjusted odds ratio (AOR) and respective 95% confidence interval (CI). Factors that were associated with outcomes at the 20% significant levels in the bivariate model were included in the multivariable analysis.

SPSS Statistic Version 28 software was used for statistical analysis.

### 2.5. Ethical Issues 

The protocol was in accordance with the declaration of Helsinki and approved by the National Bioethics Committee of Mozambique ref: 362/CNBS/20. All the variables included in this study were anonymized and no patient information was disclosed to any third person.

## 3. Results

### 3.1. Retention in Care

The characteristics of the study population according to the level of retention in care at 12 months are described in Table 1.

Around 75% of individuals were active after 12 months of follow-up, equally distributed between females and males (around 50% for each one). Regarding age, most of the participants, 32.2% (401/1247), were aged between 30 and 39 years (*p* < 0.001) and younger patients <39 years presented higher levels of dropouts (*p* < 0.001). In total, 75.0% (311/1246) of the study population lived more than 5 km away from the Health Care Facility (75.0%, 935/1246), with higher loss to follow-up for those who lived above 5 km distance (*p* < 0.001). In total, 50.9% (551/1082) of the population had completed high school. The majority of the population was employed (83.1%) and these patients presented a higher level of retention in health care (84.6% actives vs. 73.1% dropouts) (*p* < 0.001). Moreover, patients with CD4 counts < 350 cells/mm^3^ presented significantly more dropouts of follow-up compared to patients with CD4 counts > 350 cells/mm^3^ (*p* < 0.001) (Table 1).

Adjusted ORs suggested that patients younger than 39 years old were more likely to be lost to follow-up when compared to those over 50 years old (OR: 3.143, 95%CI: 1.518–6.505, *p* = 0.002 for patients between 18 and 29 yo and OR: 2.389, 95%CI: 1.397–4.087 *p* = 0.001 for patients between 30 and 39). Unemployed patients also presented a higher probability of dropouts of follow-up (OR: 1.862; 95%CI: 1.158–2.996; *p* = 0.010) compared to those who were employed. Also, patients who started treatment with a CD4 < 350 cells/mm^3^ were more likely to drop out (OR: 1.921; 95%CI: 1.255–2.938; *p* = 0.003) when compared to those with CD4 > 350 cells/mm^3^. 

Moreover, patients with lower body mass index were less likely to drop out compared to those with higher body mass index (OR: 0.539; 95%CI: 0.217–1.338; *p* = 0.183), patients with higher haemoglobin levels were less likely to drop out when compared to those with levels of haemoglobin less than 10 g/dL and those that resided more than 5 km away from the health care unit were more likely to drop out of care when compared to those that lived nearer to the hospital (OR: 1.046; 95%CI: 0.650–1.684; *p* = 0.852) (Table 2). Despite these variables being associated with loss of follow-up in the bivariate model, they did not show any association in the multivariable model.

### 3.2. Virological Response

Among HIV-positive individuals with a viral load above 1000 copies/mm^3^ (virological failure), a slightly higher number of females was observed (50.8%; 94/185), with most individuals being more than 50 years old (32.4%). Most patients (70.3%) with virological failure lived more than 5 km away from the HCF, were employed (84.7%) and presented a WHO clinical stage of 1 or 2 (82.8%) (Table 3). 

On the other hand, individuals with a viral load above 1000 copies/mm^3^ presented higher prevalence of high-level education (*p* = 0.010), higher prevalence of body mass index between 18 and 30 kg/m^2^ (*p* = 0.050), lower haemoglobin levels (*p* = 0.050) and lower CD4 counts at the start of treatment (*p* = 0.010) (Table 3).

Adjusted ORs (Table 4) suggested that patients with haemoglobin < 10 mg/L presented a higher probability of having virological failure (OR 1.810; 95%IC 1.078–3.040, *p* = 0.025) when compared to patients presenting haemoglobin levels ≥ 10 mg/L. Also, patients with CD4 counts < 350 cells/mm^3^ at the start of treatment were more likely to have virological failure (OR: 1.814; 95%CI: 1.22–2.692, *p* = 0.003) when compared to those with a CD4 count ≥350 cells/mm^3^ at the start of treatment (Table 4).

On the other hand, education, body mass index and ART regimens, which were associated with virological failure in the bivariate model, did not show any association in the multivariable model. However, it suggested that patients with higher education levels were less likely to develop virological failure (OR: 0.599 95%CI: 0.303–1.183, *p* = 0.140) when compared to patients with primary education levels, that patients with a higher body mass index were less likely to develop virological failure when compared to those with a lower body mass index (OR: 0.479 95% CI: 0.154–1.487, *p* = 0.203) and, strikingly, that patients taking regimens with a backbone of AZT + 3TC were less likely to develop virological failure (OR: 0.603 95% CI: 0.288–1.263, *p* = 0.180) when compared to those taking regimens with a backbone of TDF + 3TC.

## 4. Discussion

Herein, we aimed to describe and analyse the sociodemographic and clinical characteristics of patients being followed up at the MMH between 2016 and 2018 and their potential association with retention in care and virological failure. 

Our results showed that most of the population of patients living with HIV who started ART were retained in care for 12 months (75.1%), but some developed virological failure (16.8%). When comparing our results with previously reported levels of retention in care in Mozambique (68%) [9], we found a higher prevalence of retention in care at the end of 12 months in our HCF. This could be due to the flexibility and time schedule of patient care services and administrative and clinical staff organization, which facilitate access to health care services. 

In this study, around 25% of patients on ART were lost to follow-up at 12 months. These patients may experience higher morbidity, mortality and resistance to antiretrovirals, contributing to progression to AIDS and increasing HIV transmission, according to Stricker and colleagues (2014) [13]. However, there are some strategies that could improve retention in care, like reinforcing individual psychosocial support, understanding the educational needs of young patients and reducing time between medical appointments [14]. Also, health promotion, counselling and psychological support will only have an effect if patients are kept in the services [15].

Our results also suggest that younger age groups less than 40 years old were more likely to drop out of follow-up in the first 12 months when compared to the age group of those over 50 years old. This could be related to stigma and fear of disclosure of their HIV status to their partner [16]. These results are in accordance with a prospective observational study conducted in Los Angeles that found that younger age groups were two times more likely to drop out of treatment [17]. Other studies also showed similar results. A systematic review carried out in 2012 found that older age reduces the risk for non-adherence [18], a study in Mozambique showed that young adults (20–24 years of age) had a higher likelihood of being retained in care post-ART when compared to adolescents (15–19 years of age) [19] and another study in the United States showed that older age groups were less likely to abandon treatment [20]. However, some studies have shown opposite results, such as a study in Canada between 2006 and 2010 which showed that younger ages were less likely to drop out of treatment compared to older ages (>50 years old) [14] and in France, another study showed that the elderly were more likely to drop out of treatment, potentially due to side effects [21]. 

In our study, we also found an association between loss to follow-up and unemployment. This can be related to the socioeconomic limitations that unemployed individuals face and lower access to HCFs, related to language barriers or transportation difficulties [22]. This could happen, sometimes, because of financial difficulties that contribute to a lack of access to transportation or to daily meals, which sometimes can also lead to forgetting doses, which contributes to low adherence. This association might also be related to mobility issues, since sometimes they must move to distant areas in search of work to support their daily lives. 

Regarding CD4 cell count, our study showed that patients with low CD4 counts were more likely to drop out of follow-up, having more difficulties in improving their clinical status, suggesting that a worse health condition might be an obstacle to accessing the HCF [23]. However, we can also see it the other way round, given that these patients might have presented late to care and therefore were always at risk for lower access to care from the beginning of follow-up.

Despite there being no statistically significant differences in the distance to the health facility, we found that patients who lived far from the hospital (≥5 km) were more likely to drop out, which can be justified by the socioeconomic difficulties these patients face to reach the heath facility; often, they need to pay for transport and it takes a long time to get to the hospital. Similar results were found in a systematic review of 66 studies across 15 countries in sub-Saharan Africa where geographic and transportation-related barriers were associated with worse outcomes throughout the continuum of HIV care and treatment services [24].

Virological failure among the patients enrolled in this study was 16.8%. This might be due to the strong involvement of health care professionals in the follow-up of HIV-positive individuals. At the MMH, health care workers (i) carry out adherence counselling every day in the morning before medical appointments to raise awareness about the importance of taking the ARV regimen correctly and (ii) contact peer educators and psychologists who help patients with questions related to adherence to the ARV regimen. 

Regarding the associations between the studied variables and virological response, our results showed that patients with CD4 counts below 350 cells/mm^3^ at the beginning of treatment had the highest prevalence of virological failure. This could be related to behavioural determinants that imply that patients who present with lower CD4 counts have lower access to health care services and therefore higher risk of low adherence and worse follow-up. On the other hand, higher CD4 improves the health status of the patient, which contributes to higher access to care, better adherence, better follow-up and therefore reduction in the occurrence of virological failure [25]. Similar results were found in a systematic review involving several academic journals in 2019 [26]. The same results were reported by a case–control study in Ethiopia [27], as well as in several other studies in different parts of the world [28,29,30]. 

We also observed that patients with haemoglobin levels < 10 were more likely to develop virologic failure. Haemoglobin levels are related to disease progression in HIV-1-infected patients. Low haemoglobin levels contribute to reduced immunity, which in turn also may compromise virologic response [31]. Similar results, in which low haemoglobin levels were also found to affect treatment response, were found in two studies, one from Cameroon between 2010 and 2012 involving 951 participants [32] and the other from Uganda [33].

Although there were no statistically significant differences in multivariable models, we found that patients who completed higher-level education were more likely to develop virological failure, which may be because these patients do not trust the health system and consequently do not follow the clinic recommendation. Different results were found in a cohort study performed in Italy including 8023 HIV-positive patients that showed that patients with higher educational levels were more likely to reach an undetectable viral load compared with others [34].

While some studies show differences in virological response between age groups, our study did not find a statistically significant association. This could suggest that the effectiveness of treatment in our population is not affected by age, if it is taken regularly. Similar results to ours were found in a cohort from several African countries between 2013 and 2019 [35]. However, a study conducted in the USA between 1989 and 2006 showed that older ages had a better virological response compared to younger ones [36], and another study [21] suggested that older ages tolerated the treatment better after an intervention, raising awareness about the importance of correct adherence to treatment.

The prevalence of HIV at older ages has been increasing significantly in sub-Saharan African countries. Age can be a factor associated with a weaker virologic response, especially in older ages, but previous studies found conflicting results. Several published studies concluded that older patients were more likely to progress to virological failure and mortality, while other studies show the opposite. Some studies carried out in Africa have suggested that the mortality associated with HIV in African countries is higher in older patients compared to in high-income countries. There are also controversies regarding sex and age in relation to the virological response to antiretroviral drugs. Several pharmacological studies have described differences in pharmacokinetics, toxicity and virological response between males and females [37]. Another study conducted by euroSIDA did not find differences in virologic response between men and women [38]. Herein, we found that there were no statistically significant differences between sexes in terms of retention in care or virological response. Similar results were found in a literature review involving 64 studies carried out in several countries around the world with different levels of development [39]. The same results were found in a study in Tanzania [40] and another in the USA [41]. However, a study carried out in Mozambique between 2016 and 2021 involving 89 clinics in different parts of the country found different results, showing that women had a higher percentage of retention in care compared to men and a better virological response [19]. On the other hand, a cohort study carried out in 2016 in London involving 1190 participants showed that women were more likely to develop a poor virological response, and this was attributed to socioeconomic problems [42]. 

In summary, the results concerning the potential association of age and sex with retention in care and virological response are conflicting and should be further explored.

## 5. Conclusions

Our study indicated younger age groups (<40 years), unemployment, CD4 counts below 350 cells/mL and Hb < 10 as risk factors associated with lower retention in care and development of virological failure. Given the high rates of loss to follow-up in Mozambique, we propose investing in focused adherence reinforcement programs for patients in these risk groups. However, other studies involving a greater number of HCFs in different regions of the country are needed.

## Figures and Tables

**Figure 1 viruses-15-01978-f001:**
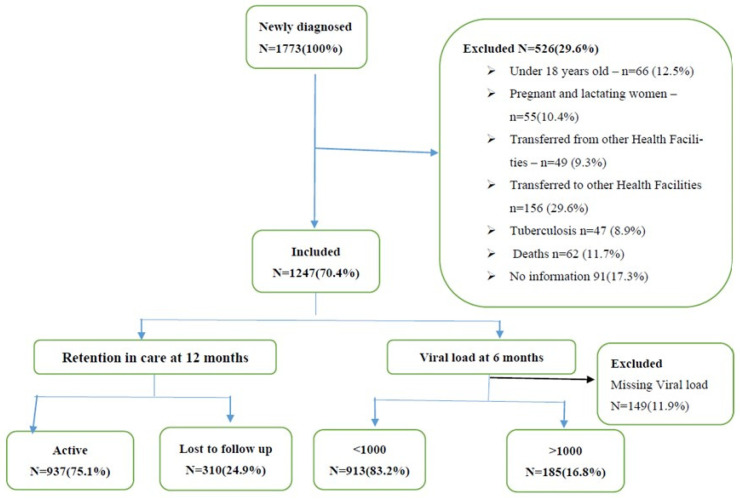
Flowchart of identification and selection of study patients, as well as their characterization in relation to retention at 12 months and viral load at 6 months.

**Table 1 viruses-15-01978-t001:** Sociodemographic and clinical characteristics of patients included in the study, stratified into actives (retained in care) and dropouts of follow-up at 12 months.

Patients Characteristics	Total	Actives	Dropouts	*p*-Value
Sex n (%)	1247 (100%)	937 (75.1%)	310 (24.9%)	
Male	617 (49.5%)	461 (49.2%)	156 (50.3%)	0.732
Female	630 (50.5%)	476 (50.8%)	154 (49.7%)	
Age at start of treatment (years), median, IQR n (%)	1247 42 (18–88)	937 43 (19–88)	310 39 (18–76)	
18–29	159 (12.8%)	101 (10.8%)	58 (18.7%)	<0.001
30–39	401 (32.2%)	286 (30.5%)	115 (37.1%)	
40–49	312 (25.0%)	241 (25.7%)	71 (22.9%)	
≥50	375 (30.1%)	309 (33.0%)	66 (21.3%)	
Residency (distance to HCF) n (%)	1246	936	310	
<5 KM	311 (25.0%)	281 (30.0%)	30 (9.7%)	<0.001
≥5 KM	935 (75.0%)	655 (70.0%)	280 (90.3%)	
Employment n (%)	1082	937	145	
Employed	899 (83.1%)	793 (84.6%)	106 (73.1%)	<0.001
Unemployed	183 (16.9%)	144 (15.4%)	39 (26.9%)	
Education n (%)	1082	937	145	
Primary school	358 (33.1%)	310 (33.1%)	48 (33.1%)	0.719
High school	551 (50.9%)	474 (50.6%)	77 (53.1%)	
Higher education (bachelor, master, PhD)	173 (16.0%)	153 (16.1%)	8 (13.8%)	
WHO clinical stage n (%)	1072	930	142	
I ou II	909 (84.8%)	789 (84.8%)	120 (84.5%)	0.918
III ou IV	163 (15.2%)	141 (15.2%)	22 (15.5%)	
Body mass index (kg/m^2^)	1063	928	135	
<18	37 (3.5%)	29 (3.1%)	8 (5.9%)	0.134
18–30	870 (81.8%)	758 (81.7%)	112 (83.0%)	
>30	156 (14.7%)	141 (15.2%)	15 (11.1%)	
Haemoglobin (g/dL)	852	739	113	
Median (IQR)	11.9 (5.2–16.9)	12 (5.2–16.9)	11.7 (5.7–15.4)	
<10	118 (13.8%)	96 (13.0%)	22 (19.5%)	0.063
≥10	734 (86.2%)	643 (87.0%)	91 (80.5%)	
CD4 baseline (cells/mm^3^), median (IQR)	1070 359 (4.0–1765.0)	934 367 (4.0–1765)	136 306 (7.0–1615)	
<350	518 (48.4%)	437 (46.8%)	81 (59.6%)	<0.001
≥350	552 (51.6%)	497 (53.2%)	55 (40.4%)	
ART regimens n (%)	1225	928	297	
TDF/3TC/EFV	1101 (89.9%)	838 (90.3%)	263 (88.6%)	0.384
AZT/3TC/EFV ou AZT/3TC/NVP	124 (10.1%)	90 (9.7%)	34 (11.4%)	

(WHO—world health organization; ART—antiretroviral therapy; TDF—tenofovir; 3TC—lamivudine; EFV—efavirenz; AZT—zidovudine; NVP—nevirapine; IQR—interquartile range: 25–75%).

**Table 2 viruses-15-01978-t002:** Determinants associated with loss to follow-up at the end of 12 months during the study period (OR: odds ratio; aOR: adjusted odds ratio; CI: confidence interval).

Active/Dropout		Unadjusted Model		Adjusted Model	
		OR (95%CI)	*p*-Value	aOR (95%CI)	*p*-Value
Sex	Female	Ref	Ref	Ref	Ref
	Male	1.046 (0.809–1.352)	0.732	1.368 (0.889–2.107)	0.154
Age at the start of treatment	18–29	2.689 (1.770–4.085)	<0.001	3.143 (1.518–6.505)	0.002
	30–39	1.883 (1.336–2.652)	<0.001	2.389 (1.397–4.087)	0.001
	40–49	1.379 (0.948–2.007)	0.093	1.665 (0.923–3.002)	0.090
	>50	Ref	Ref	Ref	Ref
Residency	<5 KM	Ref	Ref	Ref	Ref
	≥5 KM	4.004 (2.680–5.983)	<0.001	1.046 (0.650–1.684)	0.852
Employment	Employed	Ref	Ref	Ref	Ref
	Unemployed	2.026 (1.348–3.046)	0.001	1.862 (1.158–2.996)	0.010
Body mass index (kg/m^2^)	<18	Ref.	Ref	Ref	Ref
	18–30	0.536 (0.239–1.201)	0.130	0.539 (0.217–1.338)	0.183
	>30	0.386 (0.150–0.994)	0.048	0.592 (0.205–1.709)	0.333
Haemoglobin (g/dL)	<10	Ref	Ref	Ref	Ref
	≥10	0.618 (0.370–1.031)	0.065	0.649 (0.377–1.117)	0.118
CD4 at the start of treatment	<350	1.675 (1.162–2.415)	0.006	1.921 (1.255–2.938)	0.003
	≥350	Ref	Ref	Ref	Ref

**Table 3 viruses-15-01978-t003:** Characteristics of patients according to virological failure at the end of 6 months of follow-up. (TDF—tenofovir; 3TC—lamivudine; EFV—efavirenz; AZT—zidovudine; NVP—nevirapine).

Patients Characteristics	Total	VL ≤ 1000	VL > 1000	*p*-Value
Sex	1098	913 (83.2%)	185 (16.8%)	0.916
Male	544 (49.5%)	453 (49.6%)	91 (49.2%)	
Female	554 (50.5%)	460 (50.4%)	94 (50.8%)	
Age at start of treatment (years)	1098	913	185	
18–29	124 (11.3%)	97 (10.6%)	27 (14.6%)	0.393
30–39	345 (31.4%)	293 (32.1%)	52 (28.1%)	
40–49	280 (25.5%)	234 (25.6%)	46 (24.9%)	
≥50	349 (31.8%)	289 (31.7%)	58 (32.4%)	
Residency (distance to hospital)	1097	912	185	
<5 KM	310 (28.3%)	255 (28.0%)	55 (29.7%)	0.626
≥5 KM	787 (71.7%)	657 (72.0%)	130 (70.3%)	
Employment	1017	847	170	
Employed	861 (84.7%)	717 (84.7%)	144 (84.7%)	0.986
Unemployed	156 (15.3%)	130 (15.3%)	26 (15.3%)	
Education level n(%)	1017	847	170	
Primary	337 (33.1%)	283 (33.4%)	54 (31.8%)	0.001
High school	517 (50.8%)	418 (49.4%)	99 (58.2%)	
Higher	163 (16.0%)	146 (17.2%)	17 (10.3%)	
WHO clinical stage n (%)	1011	842	169	
I ou II	859 (85.0%)	719 (85.4%)	140 (82.8%)	0.397
III ou IV	152 (15.0%)	123 (14.6%)	29 (17.2%)	
Body mass index (kg/m^2^) n (%)	1003	840	163	
<18	31 (3.1%)	21 (2.5%)	10 (6.1%)	0.050
18–30	821 (81.9%)	686 (81.7%)	135 (82.8%)	
>30	151 (15.1%)	133 (15.8%)	18 (11.0%)	
Haemoglobin (g/dL) n (%)	810	679	131	
<10	106 (13.1%)	80 (11.8%)	26 (19.8%)	0.050
≥10	704 (86.9%)	599 (88.2%)	105 (80.2%)	
CD4 (cells/mm^3^) at the start of treatment, n (%)	1013	845	168	
<350	486 (48.0%)	380 (45.0%)	106 (63.1%)	0.001
≥350	527 (52.0%)	465 (55.0%)	62 (36.9%)	
ART regimens n (%)	1076	900	176	
TDF/3TC/EFV	968 (90.0%)	803 (89.2%)	165 (93.8%)	0.068
AZT/3TC/EFV ou AZT/3TC/NVP	108 (10.0%)	97 (10.8%)	11 (6.3%)	

**Table 4 viruses-15-01978-t004:** Determinants associated with virologic failure at the end of six months of follow-up.

VL < 1000/VL > 1000		Unadjusted		Adjusted Model	
		OR (95%CI)	*p*-Value	aOR (95%CI)	*p*-Value
Sex	Female	Ref	Ref	Ref.	Ref.
	Male	0.983 (0.717–1.348)	0.916	1.260 (0.842–1.885)	0.262
Education	Primary	Ref	Ref	Ref.	
	High school	1.241 (0.862–1.787)	0.245	1.247 (0.811–1.916)	0.314
	Higher	0.610 (0.341–1.090)	0.095	0.599 (0.303–1.183)	0.140
Body mass index (kg/m^2^)	<18	Ref	Ref	Ref.	
	18–30	0.413 (0.190–0.897)	0.025	0.658 (0.247–1.757)	0.404
	>30	0.284 (0.116–0.699)	0.006	0.479 (0.154–1.487)	0.203
Haemoglobin (g/dL)	<10	1.854 (1.137–3.022)	0.013	1.810 (1.078–3.040)	0.025
	≥10	Ref	Ref	Ref.	Ref.
CD4 at the start of treatment	<350	2.092 (1.487–2.943)	<0.001	1.814 (1.22–2.692)	0.003
(cells/mm^3^)	≥350	Ref	Ref	Ref	Ref
ART regimens	TDF/3TC/EFV	Ref	Ref	Ref	Ref
	AZT/3TC/EFV or AZT/3TC/NVP	0.552 (0.289–1.053)	0.071	0.603 (0.288–1.263)	0.180

(OR: odds ratio; aOR: adjusted odds ratio; CI: confidence interval).

## Data Availability

The data will be available upon justified request by e-mail to martapingarilho@ihmt.unl.pt.

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
