# Peer review of "Retention in Care and Virological Failure among Adult HIV-Positive Patients on First-Line Antiretroviral Treatment in Maputo, Mozambique"

_viruses, 2023, doi:10.3390/v15101978_

Round 1
Reviewer 1 Report
Introduction - in second and third paragraph - some repeating of stats
Methods: Rationale for exclusion criteria as It would be interesting if pregnancy/lactation and TB status has an impact on retention in care in this patient population as well?
Results:
Figure 1: Ensuring punctuation is consistent (ie periods vs commas), last box- would be nice if Missing not cut in half, missing %)
- Was VL available at month 12?
Table is suggestive that the dropouts had more people with residence close to the hospital which is different than what is stated in lines 153-154
- Appearance of Tables should be consistent (ie Table 2 is different), Table 2/4 more difficult to delineate the groupings of the rows
Discussion: Grammar/Syntax of discussion could benefit from edit
As noted above would benefit from editing of English language particularly in the discussion
Author Response
Thank your comments on the manuscript.
Please see the attachment

Reviewer 2 Report
The article by Joaquim et al. is a retrospective study of HIV-infected individuals in a health clinic in Mozambique to follow up individuals during ARV treatment. Data were analysed between 2016 and 2018. Individuals over the age of 18 were included. Of the 1773 files selected, 1247 were retained in the study. The conclusions are that clinical and socio-demographic determinants are important factors in the follow-up and virological control of individuals. The article is very clear and well written.
Comments
- In Figure 1, the box says 'transferred from other units'. This needs to be better explained, as it is not included in the non-inclusion criteria. In addition, "missing viral load" and its inclusion/non-inclusion in the analyses needs clarification.
- The justification for the 10 g/dl haemoglobin threshold is unclear. The link to reference 13 does not work and the reference is a general approach to anaemia. Furthermore, reference 33 refers to the initial haemoglobin level as an indicator of progression, but not in relation to viral load. As suggested in the article, it is recommended that haemoglobin levels be monitored to better identify individuals who are progressing and not virologically failing. This is not a single measure.
- In line 135, "In addition, patients with CD4 < 350 cells/mm3 presented significantly more loss to follow-up compared with patients with CD4 > 350 cells/mm3", it is difficult to have an idea without the incidence of mortality.
- In Table 3, in the row "Level of education", it is not easy to know which point corresponds to the significant p-value.
- On the other hand, patients with low immunity have higher rates of viral replication, compromising the immune system and leading to more frequent virological failures" I'm not sure I understand the meaning of this sentence. Most failures are related to non-adherence to ARVs.
- I do not understand the last sentence "Taking more than one pill a day (first line AZT regimen) as risk factors associated with lower retention", why target AZT?
- The article lacks cross-analyses. For example, are people who live more than 5 km away and have a higher level of education also lost to follow-up? Table 4 therefore needs better discussion.
- An important number of variables have been compared in the article and I think a statistical correction is needed for some the analyses.
Author Response
Thank you for your comments.
Please see attachemnt

Round 2
Reviewer 1 Report
Authors have incorporated edits
Reviewer 2 Report
I thank the authors for their reply.